# PROGRESSIVE PRUNE NETWORK FOR MEMORY EFFICIENT CONTINUAL LEARNING

**Joel Nicholls**
Leapmind, Inc., Japan
joel@leapmind.io

**Sakyasingha Dasgupta**
Ascent Robotics, Inc. Japan
sakya@ascent.ai

## ABSTRACT

We present a method for the transfer of knowledge between tasks in memory-constrained devices. In this setting, the per-parameter performance over multiple tasks is a critical objective. Specifically, we consider continual training and pruning of a progressive neural network. This type of multi-task network was introduced in Rusu et al. (2016a), which optimised for performance, while the number of parameters grew quadratically with the number of tasks. Our preliminary results demonstrates that it is possible to limit the parameter growth to be linear, while still achieving a performance boost, and sharing knowledge across different tasks.

## 1 INTRODUCTION

Artificial neural networks, especially their deep variants have shown remarkable success over a range of different tasks. However, deep neural networks take up significant memory, being comprised of a network structure with many thousands of parameters. Therefore, if we are to use neural networks to enable on-edge device proficiency at multiple tasks, it is necessary to design networks that can share information between multiple tasks, while having minimum memory overhead.

In this paper, we build on the architecture described in Rusu et al. (2016a) - progressive neural networks. These networks allow for transfer of information between different tasks via learnable *adapters*. Each task is solved by a sub-network called a column, while lateral adapters transfer information from old to new columns, giving them a boost in performance, while preserving the knowledge learned by the old columns. Originally used for reinforcement learning on Atari games, progressive networks have also been used for transfer from simulated to real policies (Rusu et al., 2016b) and in non-reinforcement learning tasks (Gideon et al., 2017).

However, the lateral adapters contain additional learned parameters. Therefore, the vanilla progressive neural network suffers from a quadratic increase in the number of parameters with increase in the number of tasks (Rusu et al., 2016a). Each parameter takes up memory, making the vanilla progressive network poorly-suited for use on memory-constrained devices.

## 2 METHOD

We introduce a method for continual learning with compression that limits the parameter growth to be linear with the number of tasks, while giving a performance boost using the information from previously learned tasks. In the vanilla progressive neural network, each column is trained in turn, along with the adapters that feed into the column from previous columns. In our method, each column undergoes a training-pruning-training cycle, where the adapters into the current column are also included in the pruning step. The method is outlined using pseudocode in Algorithm 1. Pruning of parameters is carried out using a simple importance measure based on their absolute value Han et al. (2015).

As successive columns benefit from the information obtained from previous tasks, the level of pruning can be controlled in order to eliminate unnecessary parameters, maintaining high accuracy for the network. The overall architecture is shown in Figure 1. Each task is highlighted as a column; later columns decrease in size with minimal reduction in performance.

**Input:** Column - deep neural network architectures $\{C_1, C_2, \ldots, C_N\}$ to be trained
**for** $i \leftarrow 1$ **to** $N$ **do**

  Import $C_k$ ($\forall k < i$) for inference
  Initialize $C_i$ and adapters $a_{k,i} : C_k \rightarrow C_i$ randomly
  `train`($C_i$)
  Prune $C_i$ and adapters $a_{k,i}$ ($\forall k < i$)
  `train`($C_i$)

**end**

**Function** `train`($C_i$)**:**

  Calc `forward`($C_i$,`forward`($C_{i-1}$,`forward`($C_{i-2}$,...)))
  Update $C_i$ and adapters $a_{k,i}$ ($\forall k < i$)

**return**

**Algorithm 1:** Progressive prune method for $N$ tasks.

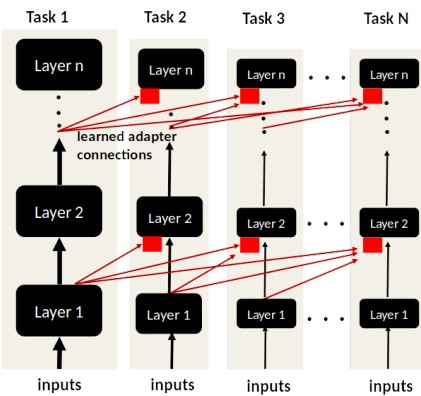

Figure 1: Overall architecture for progressive pruned network. Each column is trained in turn, on a given task. Learned lateral adapters transfer information from previous columns. While being trained in turn, each column and the adapters feeding it are pruned, removing unnecessary parameters.

## 3 EXPERIMENT

In our preliminary experiments,[1] we have used columns with 3 fully-connected weights layers for multiple classification tasks. The adapters are each made up of a multilayer perceptron with one hidden layer and a scalar multiplier to adjust the magnitude of the various lateral inputs, as explained in more detail by Rusu et al. (2016a). Our specific hyperparameters are detailed in Appendix A.

First we compare the ability of the lateral connections when transferring information between columns on the same task, with reduced successive column sizes overall (without pruning). This gives an indication for what is possible when tasks are very similar. This comparison is given in Figure 2(a), on the task of FashionMNIST classification (Xiao et al., 2017). The accuracy of successive columns increases slightly for the learned adapters, whereas no adapters or adapters that simply inject the previous column's activations cause a loss in performance. This suggests that intelligent adapters are necessary for good information transfer between tasks, although we would like these adapters to contain as few parameters as possible.

To evaluate our progressively pruned network, we use the three tasks of, MNIST (Lecun et al., 1998), (cropped-digits) SVHN (Netzer et al., 2011), and Flip-SVHN, which is created from the SVHN dataset by random vertical and horizontal flips on the input image. The third task has the difficulty of distinguishing between the digit 6 and the digit 9 rotated by $180°$ (for example), which makes it the hardest of the tasks.

---

[1]We plan to release the code for these experiments at the time of the conference

Figure 2(b) plots the mean test error over all three tasks against the total number of parameters (after pruning), for several networks. We compare our progressive pruned network to an unlinked version for several levels of pruning severity. In each case, our progressive pruned network compares favourably. To make a fair evaluation, the adapter parameters are included in the total parameter count. Therefore, at the same number of total parameters, the unlinked pruned network is allowed to have more within-column parameters than the progressive pruned network.

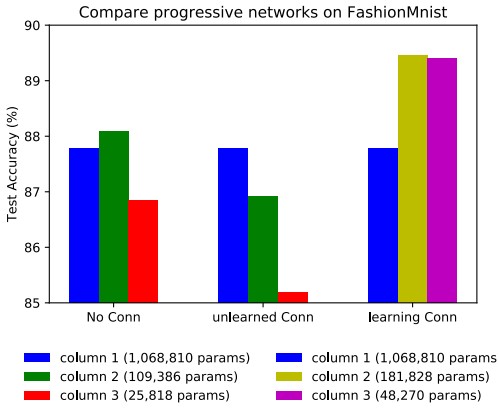 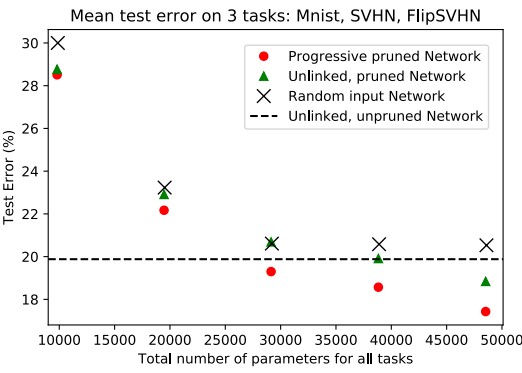

Figure 2: (a) Test accuracy of each column trained on FashionMNIST, for three different network types. No Conn is an unlinked network, unlearned Conn is linked by adapters with zero learnable parameters, and learning Conn is linked with learned adapters. In each case, successive columns are allowed fewer parameters. (b) Mean test error over the three tasks of MNIST, SVHN, and SVHNFlip, against the total number of parameters. Three network types are plotted : our progressive pruned network, an unlinked pruned network, and a pruned network where the input from lateral connections is replaced with random (white) noise. For comparison, the unlinked unpruned network has 930,270 parameters overall.

We also include a random input network for Figure 2(b), which has the same architecture and pruning method as the progressive pruned network, except the lateral inputs to each column are replaced with white noise. This network performs worst, demonstrating that the benefit of the adapter is truly due to information transfer, and not due to some kind of regularization (simply removing the adapter would not be enough to discount regularization). Thinking of the two axes as the objectives we would like to optimise, the Pareto optimal front is comprised of only the progressive pruned network.

## 4 DISCUSSION

In summary, we demonstrate progressive pruning of artificial neural networks for multiple tasks. Under a fixed budget of parameters, this allows for a boost to performance on multiple tasks. Alternatively, for the same level of performance, the number of parameters grows less quickly than for the equivalent unlinked pruned network. This is a positive step in the use of deep neural networks for multiple tasks on memory-constrained devices.

We have also made some initial experiments on convolutional networks, as opposed to fully-connected. For the set of tasks (MNIST, SVHN, and FlipSVHN), we find the convolutional progressive pruned network has no benefit over the convolutional unlinked pruned network, for an equal number of overall parameters. A possible reason for this is that the three tasks are too similar within the function space that can be represented by convolutional neural networks. The architecture of convolutional neural networks is implicitly designed to work on image data. Conversely, the fully-connected neural network has no such implicit knowledge; therefore, it benefits from a previous column that has been trained on an image based task. More generally, this could indicate that progressive pruned networks work well when different tasks have a similarity that cannot easily be exploited by redesigning the network architecture.

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

# Appendices

## A HYPERPARAMETER DETAILS

All implementations were carried out using PyTorch (Paszke et al., 2017) to run experiments using stochastic gradient descent of minibatch size 64 with learning rate 0.1 and momentum factor 0.5. Input images are normalized to the range $[-1, 1]$. For activations, we use the rectified linear unit.

For the results shown in Figure 2(a), each column is fully-connected with two hidden layers. The number of neurons for the hidden layers are 1024,256 ; 128,64 ; and 32, 16 for the 1st, 2nd, and 3rd columns respectively. For training each column, we use 10 epochs.

To obtain the results of Figure 2(b), we again use fully-connected columns with two hidden layers. The number of neurons in each of these layers is initially 128, 64 for each column. For training each column, we use 10 epochs, then prune, then train another 10 epochs. For the unpruned network, we allow for 20 epochs of training, without the pruning step. We based our PyTorch pruning algorithm on that described in the GitHub repository Wang & Ding (2017).

To get the input data into the same format, we convert the (grayscale) MNIST images to RGB images, and crop the SVHN images to 28 by 28 pixels. For the progressive pruned network, we

prune to several values of pruning fraction. For the unlinked pruned network, we simply prune to an equal number of final parameters as the progressive prune network. In both cases, pruning is done by ordering the parameters by absolute value and removing the smallest ones.

