# OpenReview forum: "Progressive prune network for memory efficient continual learning"
_ICLR.cc/2018/Workshop — Reject_

### Official Review · AnonReviewer1 · 2018-03-05
**Simple yet effective pruning of progressive networks can lead to improved generalization**

**Rating:** 5
**Confidence:** 4

**Review:**

This paper proposes a simple parameter pruning technique, based on small absolute values, as a way to reduce the quadratic growth of standard progressive neural networks to linear in the number of tasks, with subunitary coefficients.. Experiments show that, in certain cases, this simple approach delivers competitive generalization with much reduced model growth; the added columns are indeed much smaller than the first, suggesting that similar tasks can indeed leverage representations in previous columns.

Pros:
As far as I can tell the experiments are sound and interesting.

Cons:
Background work section is missing. Please add references to at least: [1, 2, 3]
Lack of details in method description, especially about pruning strategy. These should be moved to the main text.
Diagrams of the baselines considered would be useful.
Please add a table with actual numbers for each column and task. It’s not clear what accuracies over the 3 datasets really are from the average.

Please clarify:
How does your approach compare to state-of-the-art results on each dataset? Are there other ways to get similar performance within the same numbers of parameters?

[1]  A. V. Terekhov, G. Montone, and J. K. O’Regan, Knowledge Transfer in Deep Block-Modular Neural Networks. Cham: Springer Interna- tional Publishing, 2015, pp. 268–279.
[2] I. Misra, A. Shrivastava, A. Gupta, and M. Hebert, “Cross-stitch networks for multi-task learning,” in Proceedings of the IEEE Con- ference on Computer Vision and Pattern Recognition, 2016, pp. 3994– 4003.
[3] Ark Anderson, Kyle Shaffer, Artem Yankov, Court D. Corley, Nathan O. Hodas. Beyond Fine Tuning: A Modular Approach to Learning on Small Data. Arxiv https://arxiv.org/abs/1611.01714


I am willing to increase my score if substantial clarifications are made.

---

### Official Review · AnonReviewer3 · 2018-03-09
**Controlling parameter growth in neural nets for shared tasks**

**Rating:** 5
**Confidence:** 3

**Review:**

The paper presents a method for controlling the growth in the number of parameters when dealing with shared tasks in which sharing of knowledge between tasks is crucial for overall performance. It is argued that the earlier work of Rusu et al. (2016)a for achieving the same goal had a quadratic growth in the number of parameters.

In this work, the authors prune the small weights based on their magnitudes as done is Han etal. 2015. In this sense, there is limited novelty in the paper. Also, it is not clear how the parameter are linear in the number of tasks after pruning as claimed by the authors. These still seem to be quadratic wrt number of tasks over the reduced set of parameters after pruning.

---

### Official Review · AnonReviewer2 · 2018-03-10
**Very preliminary work, claims not properly substantiated.**

**Rating:** 5
**Confidence:** 3

**Review:**

Building on progressive NNs for multi-task learning, this work aims at reducing the number of weights (and therefore memory) used by the solution. The proposal is simply to use a previously existing and straightforward pruning mechanism on the weights, so it doesn't convey a particularly novel idea.

Progressive NNs suffer from a quadratic increase in the number of weights with the number of tasks. In this paper, the authors claim that that memory increase is only linear. However this is not justified, since the number of adaptor networks is still quadratic. No reason is given for this linear order. One possibility is that the authors are adjusting the pruning level in a way that achieves the desired scaling, but then we might as well say that it is sublinear, superlinear, etc, so that the linear scaling is not a property of the approach

In Figure 2(a) it seems that "No conn" networks result in better accuracy with the intermediate number of parameters, meaning that the first network is overfitting. This is not the regime in which the test should have been done.

Figure 2(b) doesn't include a comparison with the progressive unpruned networks, which should be the reference.

---

### Decision · Program_Chairs · 2018-03-20
**ICLR 2018 Workshop Acceptance Decision**

**Decision:**

Reject

**Comment:**

Based on the reviews, this paper has not been accepted for presentation at the ICLR workshop. However, the conversation and updates can continue to appear here on OpenReview.